# Adolescent and Parent Experiences of Acceptance and Commitment Therapy for Pediatric Chronic Pain: An Interpretative Phenomenological Analysis

**DOI:** 10.3390/children6090101

**Published:** 2019-09-07

**Authors:** Marie Kanstrup, Abbie Jordan, Mike K. Kemani

**Affiliations:** 1Department of Clinical Neuroscience (CNS), K8, Psychology, Karolinska Institutet, 171 77 Stockholm, Sweden; 2Functional Area Medical Psychology, Karolinska University Hospital, 171 76 Stockholm, Sweden; 3Centre for Pain Research, University of Bath, Bath, BA2 7AY, UK; 4Department of Psychology, University of Bath, Bath, BA2 7AY, UK

**Keywords:** pediatric chronic pain, Acceptance and Commitment Therapy (ACT), Interpretative Phenomenological Analysis (IPA), lived experience, values, children, adolescents, parents, experiences of treatment

## Abstract

Pediatric chronic pain is common and can be related to reduced functioning in many domains for the young person and their parents. Existing psychological treatments such as Acceptance and Commitment Therapy (ACT) have shown to be effective, but improvements are needed. Qualitative approaches can help improve our understanding of treatment processes and outcomes. The aim of the present qualitative interview study was to explore the lived experiences of young people and parents who had participated in ACT for pediatric chronic pain. Four young persons and four parents were interviewed, and data was analyzed using Interpretative Phenomenological Analysis (IPA). Three themes were generated, each comprising two subthemes: (1) ‘Warning system’, which included experiences from being offered this psychological intervention, and the alternative explanations provided for pain; (2) ‘Change and challenges’, which suggested the importance of the values-based work, and of individual adaptation; and (3) ’A common language’ in which the interaction with others and new ways to communicate around the pain experience were described. Findings highlight the importance of pain education, formulating and acting in line with personal values, and communication around the pain experience, as well as the need for developmental and individual adaptations of interventions.

## 1. Introduction

Chronic pain is common in young people, reported by more than one in four individuals and for a substantial number, pain has severe consequences on their emotional, social and physical functioning [1,2,3]. In qualitative studies, young people have described that their experiences of chronic pain include difficult encounters with medical professionals, limitations in functioning due to pain, extreme experiences of pain, and altered emotional wellbeing and social impairment [4,5,6,7]. Pediatric chronic pain has also been described as having an impact on identity development [8], and development and maintenance of peer relationships [9], and is related to impaired family functioning, caregiver burden and extensive health care costs [10,11,12]. 

In addition, parents of young people with chronic pain report emotional distress [13,14,15,16], helplessness, and an altered experience of parenting that is more akin to parenting a much younger child [17,18]. Thus, there is a great need for effective treatments for pediatric chronic pain that address both child and parental functioning and behavior.

Research supports the use of face-to-face psychological therapies and interdisciplinary intensive treatment to reduce pain and restore functioning in pediatric chronic pain [19,20] and parent support is commonly included in treatments, with beneficial effects [21]. However, these evidence-based treatments are not effective for all children, and some domains of pain-related dysfunction generally do not improve. For example, two studies report that 22%–27% of children show no response after intensive interdisciplinary treatment [22,23], and generally, positive effects on depression and anxiety are lacking for psychological therapies [19]. A clearer understanding of the processes and mechanisms that are related to successful outcomes for all domains of functioning, and the active ingredients of treatment, would advance our understanding of how to improve treatment outcomes [19].

Acceptance and Commitment Therapy (ACT) has shown promising results in the treatment of pediatric chronic pain, both when delivered as a weekly outpatient treatment and as an intensive interdisciplinary treatment program (see e.g., [24,25,26,27]). ACT is explicitly linked to Contextual Behavioral Science (CBS) and Relational Frame Theory (RFT), which operationalizes cognition and language primarily as verbal behavior, i.e., derived conditioning [28,29]. In derived conditioning, in contrast to respondent and operant conditioning, stimuli (objects, words, thoughts, etc.) receive their functions (e.g., fear) indirectly via and according to their specific relations to other stimuli [29]. Based on this approach, specific interventions in ACT aim to undermine problematic functions of cognitions and language, for example, by means of acceptance (i.e., noting experiences of, e.g., pain/distress, without attempting to change them) and perspective-taking exercises (i.e., placing yourself as an active observer of such thoughts/experiences), and to facilitate adaptive and flexible behavioral repertoires in line with motivating verbal antecedents (i.e., personal values) [30,31]. Following ACT for pediatric chronic pain including parental support, results show improvements in parental depressive symptoms and psychological flexibility [24,26,32], and sustained improvements in parent responses to child pain have been reported after ACT-based parent support [33]. To date, only a few studies have sought to explore factors in treatment that are related to changes in outcomes after ACT for pediatric chronic pain. Preliminary results indicate that improvements in behavioral inflexibility during treatment mediates improvements in pain-related disability after treatment [34], that changes in child acceptance of pain during ACT are associated with changes in child distress and functioning at follow-up [27], and that changes during and following ACT in parents’ acceptance of their child’s pain are related to changes in the child’s acceptance of their pain [26]. 

To increase the understanding of processes of change and outcomes in psychological treatments for young people, as well as to improve clinical care with respect to the preferences and needs of young people and parents, rigorous qualitative studies provide an important source of information [35,36,37]. Further, as described above, a few studies indicate that central treatment targets in ACT (e.g., acceptance) are related to improvements in outcomes for young people with chronic pain. However, no study has yet, to our knowledge, applied qualitative methods to explore participant experiences of ACT for pediatric chronic pain, and what may have been perceived as helpful to achieve change for participants (e.g., do participant experiences include or relate to the explicit targets of ACT, and the processes thought to facilitate change). Thus, the aim of this cross-sectional qualitative study was to explore the lived experiences of young people and parents with regard to participating in ACT for pediatric chronic pain. 

## 2. Materials and Methods

### 2.1. Study Setting and Treatment Description

This study took place in a tertiary care pain treatment service at a university hospital in Sweden. Patients eligible for referral to the services experienced longstanding disabling pain conditions. In brief, the standard outpatient ACT treatment comprised ca. 18 weekly sessions. The treatment could be delivered either as individual treatment or in a group format. The majority of sessions were led by a psychologist and specifically provided for the young people. Sessions included applying behavior analysis to pain-related situations, focusing on short-term problematic-symptom-reducing behavior, as well as on alternative values-consistent behavior; ACT-strategies such as acceptance and perspective-taking, in order to facilitate exposure to previously avoided valued activities; and communicative strategies to involve family and friends to validate experiences and support behavior change. Parent sessions included behavior analysis of problematic parent behaviors (i.e., excessive short-term symptom-reducing behaviors); clarification of parental values; using acceptance and perspective-taking to handle personal worry and distress; and how to effectively coach the child to change their behavior and function better in the presence of child pain (e.g., by means of active listening, validation of child experiences and facilitation of values-consistent behaviors). Both parents and children received pain education sessions, delivered by a physician or a nurse. These sessions included descriptions of the pain system, the differences between acute and chronic pain, and of how pain perception is context-dependent, with the aim of facilitating a shift in perspective from symptom reduction to acceptance of pain and valued living. Specific medical evaluations or adjustments in medical treatments, along with physiotherapy sessions, were available as treatment add-ons if needed.

### 2.2. Study Population, Sampling and Recruitment

Study participants comprised young people and parents. Inclusion criteria for young people were: Being aged between 11–18 years; recent participation in the standard individual or group-based ACT treatment provided at the above clinic; being able to verbally communicate their experiences in an interview. Inclusion criteria for parents comprised recent participation in the standard parent support program offered at the service for parents of patients with pediatric chronic pain.

Participants were recruited through purposeful sampling. During the final session or follow-up appointment, clinicians not involved in the study asked potential participants if they were interested in receiving additional information about the interview study from a researcher. Clinicians had been prompted by the researchers to ask both participants who they thought had responded well to treatment and were satisfied, and those who were not. If potential participants agreed to be contacted about the study, M.K. phoned them and provided study information. This included that participation was entirely voluntary and that the decision to take part or not would not affect their care in any way. After this phone call, those considering study participation received detailed written study information together with consent forms per post. This included information that any quotes used from participants would be presented in a way that would protect their anonymity. Young people and parents were able to participate individually or as dyads. If written consent was provided, interviews were scheduled and performed separately with parents and children. Eleven families agreed to receive information about the study, but six of these could either not be reached or declined to participate due to time constraints. The study was conducted in accordance with the Declaration of Helsinki, and approved by the Regional Ethical Review Board in Stockholm (dnr: 2017/2214-31).

### 2.3. Participants

A total of four adolescents and four parents were interviewed. Participants comprised three dyads, one young person (without parent) and one parent (without their child). Young people were aged 12 to 18 years (M_age_ = 16.0 years, all female) and pain duration ranged from 2 to 18 years (M_pain duration_ = 8.75 years). Their pain locations included head, neck, shoulders, abdomen, and widespread pain, and three of four young participants reported more than one pain location. Examples of pain-related impairment in functioning prior to treatment included limitations in physical activity, extensive school absence (e.g., only present 50% over the course of a year, or staying at home full-time for months at a time), pain episodes that included fear of dying or being unable to walk, numerous health-care visits, and emotional and social impairment (e.g., depressive symptoms, fear of pain, and feeling isolated and different from others). Parent participants were between 45 to 52 years (M_age_ = 48.25 years, three mothers and one father). No other parent characteristics (e.g., presence of parent chronic pain) were collected in this study. Parents described pain-related family dysfunction and difficulties in parenting prior to treatment. These included major adaptations in family activities and routines to reduce the child’s pain, constantly being on call for their child, and feeling guilty when they had encouraged everyday activities or initiated medical examinations that provoked pain. Participants had received the same treatment content, either as part of a group (*N* = 6) or in individual treatment (*N* = 2), but the treatment provider varied between the participants (four different psychologists delivered the psychologist sessions, and two different physicians and one nurse delivered the pain education component). The young people received 9–12 sessions (M_sessions_ = 10.8) with the psychologist, and 2–3 sessions with the physician. They did not have additional physiotherapy or medical evaluation sessions. The parents received three sessions with the psychologist and two sessions with the physician. The mean time between end of treatment and their participation in the interview was 11.8 weeks (Weeks_min–max_: 8.3–22.1).

### 2.4. Analytical Approach

Interpretative Phenomenological Analysis (IPA) was chosen a priori because of its particular focus on the personal lived experience [38]. IPA has been used extensively to examine a variety of health-related topics, and more specifically within psychology and the field of chronic pain [8,18,39,40,41,42,43]. IPA offers a structured and idiographic approach in which both the participant and the researchers strive to make sense of their experiences, positioning the role of the researcher as important within the analytical process. IPA adopts a phenomenological perspective, is suited to novel areas [44] and for studying pain-related phenomena [43], and typically incorporates a small sample size to reflect this focus on the detailed exploration of participants’ lived experience [45]. The selected sample size was informed by the specific idiographic focus of IPA, with its typical use of small sample sizes alongside the quality and richness of the data [38,46]. No attempt was made to achieve saturation [47]. 

### 2.5. Data Collection 

The semi-structured interview schedule was developed by M.K. with support from A.J. and M.K.K., following Smith, Flowers and Larkin [38]. Two major topics were defined: (1) The nature of life for the participants prior to participation in ACT; and (2) participant expectations and experiences of ACT-based treatment. The first topic was primarily aimed to develop rapport and contextualize their experiences. For the present study, data from topic 2 was analyzed. Additionally, to inform a potential subsequent study, participants—at the end of the interview—were asked about their experiences from participating in routine quantitative treatment evaluation using self-report questionnaires. 

Topics were explored using open-ended questions, followed by prompts if/when needed (e.g., ‘Tell me more’, ‘How did that feel’) and returning to topics that needed further exploring. A flexible adherence to schedule was employed to allow the participant to lead. M.K. (PhD, licensed clinical psychologist) conducted all interviews. She deliberately refrained from prompting or asking participants about specific content, i.e., by naming specific treatment content before the participants did. When interviewing children, professional experience from working with children is helpful for the modifications that may be needed during data collection [45], and although IPA interviews, generally, are non-guided, children may need a bit more guidance and probing. For this, M.K. drew on her clinical experience (>8 years) from working with pediatric patients.

Interviews were conducted between February 2018 and November 2018. Interviews with young people were between 48 to 78 minutes (M_time_ = 58 min) and parent interviews between 48–73 min (M_time_ = 61 min). All participants received a cinema ticket after the interview. Interviews were audio-recorded and then transferred to an encrypted hard disk drive before being transcribed verbatim by M.K. Reflective discussion took place regularly between M.K. and M.K.K. during the data collection period.

The analysis followed the processes and strategies for IPA outlined by Smith, Flowers and Larkin [38]. The first case was examined in detail, before moving on to the next. Transcripts were read and re-read by M.K. and discussed with M.K.K., and initial noting was conducted in the margin of each transcript. Then, emergent themes were developed from the notes, and connections sought between themes, before moving on to the next case and finally, patterns across cases were explored. After the initial noting, the computer program NVivo version 12 [48] was used for each of the subsequent steps of analysis. Each participant was treated as “a case on its own terms, to do justice to its own individuality” (p. 100, [38]). All participant quotations were translated by M.K., with the help from an independent professional translator. Presented quotations were sampled from all participants. 

Striving for member check, participants were provided a summary of a preliminary Swedish results section via post, with the option to send written comments back to M.K. or to schedule a time for a telephone discussion. One parent and one adolescent replied via post. The parent did not suggest any changes to the results, but the adolescent, in addition to expressing her appreciation for the study, wrote that the aim of the treatment should be made more clear when beginning treatment, i.e., that the treatment aimed to facilitate a shift from trying to get rid of pain, to instead managing the problem. She also suggested that this could be made clearer in the results theme 1 (see Results). We followed her advice on this by making sure the aim of the treatment is clearly reflected in theme 1, in addition to the discussion.

In addition to adopting an established method for interviews and analysis, we strived to enhance the quality of the study in line with several suggestions in the literature [38,49,50]. M.K. and M.K.K. both had >8 years of clinical and research experience of pediatric chronic pain, and A.J had extensive research expertise in pediatric chronic pain and IPA, all of which added to the contextual understanding and sensitivity. Participants were informed that the aim of the study was for M.K. to listen to their experiences of participating in treatment, in order to gain a better understanding of how the treatment could be improved. She stated clearly that there were no right or wrong answers and informed them that she had prior experience as a therapist using ACT at the clinic, but that she currently worked solely as a researcher at the university. She shared her personal interest in doing this study, which stemmed from prior clinical encounters with patients who had shared their experience of treatment with her, and that she now wanted to use the unique knowledge that comes from lived experiences of young people and parents in a systematic way to improve treatments. Use of an audio recorder, transcripts, and the computer program NVivo secured traceability and reproducibility of the analyses. Debriefing between the researchers took place during in-person meetings and Skype-supervision.

## 3. Results

Three superordinate themes were generated from the data by the researchers: ‘Warning system’ (superordinate theme 1); ‘Change and challenges’ (superordinate theme 2); and ‘A common language’ (superordinate theme 3). All three superordinate themes each comprised two subthemes. Pseudonyms have been used throughout the results section to ensure the anonymity of study participants.

### 3.1. ‘Warning System’ (Superordinate Theme 1)

This theme represents the ways in which participants’ understandings of pain and its meaning changed throughout the treatment process, suggesting a developmental process. Additionally, in this theme, the individual context of such beliefs about the cause of pain are explored.

#### 3.1.1. Subtheme 1. Buying into a Psychological Approach 

Coming into ACT-treatment, participants largely understood pain from a biomedical perspective, in which pain symptoms represented potential danger, underlying damage or a discoverable disease. In addition to avoidance of everyday pain-evoking activities, adopting this perspective motivated families to seek numerous medical consultations and treatments. Possibly, as these efforts to alleviate pain did not succeed, some participants were open to a different approach, focusing on the role of psychology in treatment. As described by Wilma: *“I knew you [psychologists] weren’t people who can like wave a magic wand, I knew that a lot of it would be in here (points to head), that it’s more psychological and all that” (Wilma, young person).* Whilst Wilma did not expect to be a passive recipient of a miraculous cure, she had clearly underestimated the active role that participants were expected to take in their own treatment. *“I hadn’t expected it to be so much me working with my own brain myself, that it would be me working everything.” (Wilma, young person).* This appears to contrast previous treatment experiences in which she had played a more passive role. For others, the psychological approach did not seem a realistic alternative to more medically based interventions. *“First I thought it was really stupid, and ridiculous, that it can’t get better just by talking, it just seems impossible, like” (Emma, young person)*. 

For some, even though skeptical, the decision to try a more psychological approach was motivated by the slight possibility that treatment would be helpful with regard to reducing pain and associated disability. Although such a possibility was perceived as minimal, the glimmer of hope meant that for participants such as Julia, it was impossible to pass up on this opportunity. *“I mean I’ll try anything, if I’ve been pretty skeptical about this, but I reckon if there’s anything in it that could help well it’s silly not to have a go.” (Julia, young person)*. Interestingly, some parents adopted a more flexible approach in considering psychological approaches compared to their child. For example, Lars describes that when presented with the objectives of a psychologically focused ACT treatment, he was able to appreciate and ‘buy’ into the principles of treatment whilst he describes how these same principles made little sense to his daughter.


*We talked about it afterwards and we felt that we had a different experience of that meeting she and I, I thought it was a good meeting and she [daughter with chronic pain] thought she didn’t get it at all, I guess she got frustrated as well when they told her ‘You’ll always be in pain and it’s uh up to you how you’re going to manage it’ and that wasn’t something she wanted to hear, she wanted to, she wanted to see someone who can help her get rid of the pain. (Lars, parent)*


In other instances, it was the child who was willing to embrace a more psychological approach to treatment and let go of bio-medically based explanations and treatments. For example, Elin described how, over time, she was able to appreciate a change in focus from alleviating pain, to living well with pain. *“After so many doctor’s appointments you end up so focused on just getting rid of it, and now it’s more like ‘You’re still going to feel pain and it’ll still hurt when you do stuff but it’s ok.” (Elin, young person).* In contrast, her parent was reluctant to embrace a psychologically focused treatment, expressing “*massive doubts*” about treatment effectiveness, as the parent was unsure that they had “*really tried everything” (parent pseudonym left out to prevent linking the dyad)*. This suggests that the psychological ACT treatment was viewed as a last-ditch treatment, something only to be embraced when all options to provide a biomedical approach to explaining and treating pain had been exhausted. What is interesting here is that individuals seem to reach this point of exhausting all other options at different times. Parents and young people who come to pre-treatment assessments with differing conceptualizations of the pain problem and how it ought to be solved can add complexity to the decision about whether to enter treatment or not, both in terms of actual decision making from the role of the parent as the guardian, and the motivation for engaging in treatment. It suggests the importance of individually addressing readiness to adopt the treatment in parents and young people, discuss and validate any inconsistencies, and tailor subsequent interventions thereafter.

#### 3.1.2. Subtheme 2. Reframing the Experience of Pain

Pain education, as part of ACT, aimed to provide an alternative understanding of pain and was received positively by some young people and parents. It facilitated a new understanding of how longstanding pain could be a product of faulty pain signaling, providing a clear contrast to previously held understandings of pain as solely arising as a direct consequence of damage or disease. *“I liked it when they explained why it hurts, and it’s like, oh I get it, it’s this thing right uh, it’s the warning system going off without anything really being wrong with us” (Wilma, young person)*. Some parents also expressed frustration with themselves that they had not previously appreciated the importance of the role of psychological factors in pain, expressing both relief and frustration when non-biomedical explanations of pain were provided in treatment.


*So many lightbulb moments… I’ve worked in healthcare myself and I’m familiar with psychosomatic problems but I’d never made the connection with my own child because of the signals and the symptoms and it was just like, ‘My God,’ it was so, so crushingly obvious. (Maria, parent)*


Pain education provided important confirmation of pain as a real entity and an understanding of how the pain experience is shaped by external and internal contextual factors, such as worry and the behaviors of family members. This helped parents to appreciate the active role they can play in supporting their child to manage their pain.


*Especially in and around this idea that there are so many more things that affect the pain and I think that was a great thing for our family to kind of bear in mind that when we are worried or stressed or do things, that makes it worse too. (Eva, parent)*


However, some young people expressed that the pain education was too technical and difficult to understand if you were not “*super smart” (Emma, young person).* Such difficulty also seemed to arise from the extensive use of sometimes confusing metaphors.


*He [clinician], like, goes on about, like, you know, talking in metaphors so you end up, like, ‘Do you even know what you’re talking about? (Julia, young person)*


For Julia, the use of metaphors was further problematic as their use caused offence. Here, she describes her experiences of two metaphors used in session, ‘the princess and the pea’ (from the fairy tale by Hans Christian Andersen), and an image of a man lying on a bed of nails, which had been used to illustrate individual differences in sensitivity.


*He [clinician] like compared us to the princess and the pea, so, like, she’s asleep here, right, and she’s really sensitive, when someone else wouldn’t have ever even felt it. Right, ok. And check out this fakir, he can train his pain to lie down on a bed of spikes, yeah? Like this? Like congratulations mate, but we don’t get to choose when to be in pain, he can decide how long he’s gonna stay on those spikes at a time, we can’t. We’re on those spikes all the time, no break. It didn’t add up. (Julia, young person)*


Describing these metaphors as *“degrading”* and *“condescending”,* Julia’s comments illustrate how the use of these metaphors alienated her from, rather than engaged her with, pain education. This further highlights potential pit falls in using off-the-shelf metaphors and the importance of adapting metaphors to the young person and their own particular situation.

### 3.2. Change and Challenges (Superordinate Theme 2)

There seemed to be consensus among participants that a focus of treatment was to provide strategies to deal with pain and related distress in ways that made valued activities and behaviors possible. However, opinions diverged on the specific parts of treatment that had been helpful in promoting such change. Participants repeatedly spoke of the values component of the treatment, underlining this aspect of treatment as specifically impactful, yet other ACT exercises and treatment content were perceived differently, implying potential challenges that may be addressed by tailoring treatment to participant needs.

#### 3.2.1. Subtheme 1. Values—Doing What Matters to You

The values-based work seemed to appeal to most participants, by serving as a motivational factor and as a reminder of long-term functional and valued behaviors and activities. This is illustrated by Wilma who describes how useful it was for her to write down what she valued.


*We got to write down what we think is important in our lives and what we want to achieve, or what we want to fight for, and I’m a bit of a dreamer if I’m honest, and that’s why I enjoy getting to express it and really get to see it in black and white and, it’s these things right here that really matter to me, and it has been a bit easier to, those days when it’s really hard, or at least at first, to if I thought about it, or saw it or something like that, then it was like ‘Well, **I** do want to get there, **I** want to be able to make it over there, and that means **I** have to get up, **I** have to keep going,’ that’s why I liked it a lot. (Wilma, young person)*


In the quotation above, the ‘I’ (bolded for the reader) signifies the active role of the individual, moving from a focus on reducing pain to a sense of living well with pain. This is congruent with a key tenet of ACT, to broaden the behavioral repertoire in the presence of persistent distress (e.g., pain), when control and reduction of distress is not working. Parents also described how the values work facilitated useful discussions around identified differences in parenting styles and how such differences and strengths could be used in a complementary way to support their child. For Maria, discussions around values became *“the starting point”* for her and her husband, uniting them in a newly shared approach to parenting. 


*So, it [values work] was, an opening, a way in, definitely, it was so valuable for us, um, cause it became the foundation for the conversations we have today, no question, and we used to have different points of view when it came to that. (Maria, parent).*


Formulating personal values provided clarity and direction for participants, with one mother describing how the values-based work had implications for life beyond pain, influencing her choices about what type of parent she would like to be, and other life choices, not only in the context of pediatric chronic pain. *“I mean it helps us grownups too, you know, to watch (laughs), ‘What do I want in life, and what do we want to prioritize, what do I actually enjoy.” (Eva, parent).* She further described how the values work served as a reminder to stay on track also when facing stressors in life. *“To bear in mind that you really can stay on course pretty well even though there are distractions all around. Um I thought those bits were really good, yeah” (Eva, parent).* This is important as such values-based action is an important treatment outcome in ACT: Engaging in long term effective parenting behaviors in line with parent values, also in the presence of distractions and own distress. Further, engaging in values-consistent behavior can positions parents as role models for their children.

#### 3.2.2. Subtheme 2. ‘Getting the Idea’—The Need for Adaptation

With the exception of the values-component of treatment, there was substantial variability in how participants perceived the function and the form of other ACT exercises, and although potentially challenging, this implies the need for careful tailoring of treatment to the individual. The variety of exercises was appreciated, suggesting the importance of flexibility in which exercises to use.


*I was so pleased that there was a choice of different exercises, that it wasn’t like everyone has to do this one exercise, or has to keep doing the same exercise even if it’s not working, that there was, you know, if it’s not working for you, you can try this other one instead, I liked that. (Wilma, young person)*


Clarity of the purpose of the exercise was an important factor for young people with regard to their willingness to engage with treatment. Interestingly, this appeared to be the case even when the specific elements of the exercise seemed less desirable, or were challenging to complete in everyday life. For example, Elin described how whilst it was difficult to complete a task such as saying thoughts out loud to yourself, she was able to successfully apply versions of this task, capturing the core idea of these exercises. *“But I can still get the idea and the way of thinking that you should think ‘I’m in pain AND I can still go” (Elin, young person).* For Elin, the purpose remained clear and this was the important factor in terms of encouraging engagement in such activities outside of the therapeutic sessions. Some exercises were considered difficult for young people, in that the purpose of the exercises was unclear to the participants, or that the exercises were overly demanding. This suggests the need to consider developmental factors to enable young people and parents to successfully engage in these activities. For example, Wilma found it difficult to focus on aspects of perspective-taking in the heat of the moment (noting her thoughts as something she has, i.e., ‘I have a thought’, instead of getting absorbed in the content and emotional qualities of her thoughts).


*The ‘I Have a Thought’ one, it felt like it’s for older people rather than younger people, cause us teenagers, we’re not really, we have a lot of emotions, we’re a bit, well, we’re a bit of a mess you know what I mean, and I think you need to be a bit older and a bit more well not mature exactly but a bit more together to be able to keep your head straight and think about things, but that’s the thing, for us, it kind of gets too much, for us. (Wilma, young person)*


Adopting Wilma’s perspective, normative adolescence is a period where cognitive control is still developing, and during which there is variability in emotion regulation [51]. Wilma’s difficulty in engaging with this particular exercise suggests a need to enhance the focus on emotion-regulation, by providing developmentally appropriate strategies in order to increase treatment participation and potentially, treatment effects. 

Such a variability in developmental functioning and preferences in adolescence was noted in response to other ACT exercises such as use of the ‘pain monster’ metaphor [52]. For Julia, this activity was perceived positively and enabled her to adopt a new strategy to “*distance yourself from the pain. (Julia, young person).* Yet for others, this activity was perceived as developmentally inappropriate, with Emma commenting that “*it felt a bit babyish to think like there’s a monster living in your head like the way you’d tell a child, to make it easier to understand” (Emma, young person).* Such a disparity in experience illustrates the complexity of targeting ACT-based exercises to a developing adolescent population, whose levels of cognitive and social functioning vary more widely than in an adult population. Delivering ACT-based treatment in a group format likely further increases this challenge. On a related note, none of the parents described any difficulties with understanding exercises included in the parent support sessions.

Parent treatment participation enabled parents to combine new and existing knowledge into an enhanced understanding of their child’s pain and how it can be better managed with the help from ACT strategies. Maria described this as: *“that information or training we got as parents too, and that made, I suppose that was when finally all the pieces of the puzzle fell into place, and you understood that ‘This is what’s really going on” (Maria, parent).* Further, she thought that even more focus on the role of the family would have been helpful.


*In hindsight, now that I can see that (the child’s) stomach problems have been largely psychosomatic I suppose I feel that it would have been very helpful to have looked more closely at the family situation, how mum and dad are doing, how siblings are doing, actually. (Maria, parent).*


The individual nature of parental engagement with treatment is further highlighted by Lars who was opposed to the focus on the parent’s own feelings, and described how he *“couldn’t quite see the point in us parents having, undergoing treatment which I suppose is what it felt like at the time” (Lars, parent).*

Thus, flexible adaptation should also be considered for the extent of treatment, as well as for treatment content. 

### 3.3. ‘A Common Language’ (Superordinate Theme 3)

All participants reflected on the nature and function of the communication between themselves, treatment providers and other young people and parents during treatment. This superordinate theme comprises the sub-themes: ‘Thinking outside the bubble’ and ‘A new dialogue’. 

#### 3.3.1. Subtheme 1: Thinking outside the Bubble 

The majority of participants had received treatment as part of a group (*N* = 6), a common mode of treatment delivery for many pain treatments. Overall, participation in group treatment was described positively, and provided opportunities for normalization and sharing of experiences. Being in a group, in and of itself, seemed to provide additional benefit to the specific treatment content, suggesting benefits associated with shared group membership for both parents and young people. For Julia, meeting other young people with similar problems enabled her to reflect on *“how different the same experience can be” (Julia, young person),* for example, that pain in another bodily location was still related to the same functional impact for other young people. Meeting others in a group setting also extended Julia’s knowledge about pain and her ability to see beyond her own personal challenges. Specifically, experience of receipt of treatment in a group setting provided Julia with a wider understanding of the impact on others, enabling her to appreciate the importance of individual differences with regard to pain experience and impact.


*And that you just get, cause you’re thinking inside your own little bubble, you think that if someone’s in pain then they feel what I feel, because that’s all I know, and to get to look outside the little box a bit, I think that’s bloody important. (Julia, young person)*


Focusing outwards, on other young people’s experiences, as well and not just your own struggle, may facilitate a distance to the pain experience, as well as provide a shared sense of understanding. Group membership also helped to alleviate a sense of isolation through normalizing of experiences and thoughts. This was the case for both young people and parents, with Anna (parent), describing how attending the group enabled her to gain a sense of shared perspective among fellow parents. 


*The other mother I met contributed a lot because your own thoughts go a bit crazy if you’re sick with worry and it’s a nice feeling to have it confirmed, that you’re not, so what it is at this point where it is now that um these thoughts you have, they’re um part of the situation you’re in somehow. (Anna, parent)*


But this sense of shared experience was not entirely positive for all participants. Comparing situations could also lead to a perceived sense of guilt if individuals perceived others to display greater pain and associated disability than them. Elin describes how:


*I guess it’s both good and bad to have a group, you know, it’s like you can relate but at the same time it’s like at least in my case I can sometimes feel like ‘you’re much worse off than me’. (Elin, young person)*


Perceiving a disparity in levels of functioning among group participants was uncomfortable for Elin and also resulted in her questioning her own motivations and abilities. For example, Elin began to compare herself with other young people in the group whom she perceived as more severely affected by pain than her, leading her to question her right to be present in the group. She asked herself *“what am I doing here, what am I complaining about” (Elin, young person)*. 

Perhaps more concerning were instances where engagement in a group-based treatment resulted in a sense of diminished rather than enhanced communication between group participants. This occurred in some cases due to heterogeneity in symptom burden and functioning, and due to discomfort arising from discussions of sensitive subjects for certain participants. For example, Wilma reported feeling upset about a comment made in session by a fellow adolescent who commented that he/she could not understand people with mental health problems. This resulted in a decrease in group engagement on her part.


*That made it hard, I mean, that’s when I started closing myself off a bit, er that’s not what you’re meant to do, you’re supposed to open up to the others in there, but I didn’t feel comfortable after that. (Wilma, young person)*


As Wilma noted, this is at odds with the very premise of group-based therapy in general (not just ACT), that is, open sharing of your thoughts and feelings with the group is encouraged. She felt that it may have helped her if the treatment provider had responded to this situation differently. She concluded that one-on-one treatment probably would have been better for her, again highlighting the importance of meeting the individual needs of patients when considering appropriate treatment and provision of such treatment.

#### 3.3.2. Subtheme 2: A New Dialogue 

The treatment provided new and different ways to communicate about pain and its impact on individuals’ lives. Communication was central to both the parent–adolescent interaction, and the interaction with the treatment provider. The opportunity to convey her suffering to someone who validated her experience had helped this young participant.


*It feels so good that someone else gets it too, that, like, yes you’re in a shitload of pain and there’s nothing I can do right now to make you hurt less but I can at least listen, and that feels good. (Julia, young person)*


Emma noted that *“I felt I could be open with her [clinical psychologist] and that felt good” (Emma, young person).* For her, this was in contrast to her previous encounters with health care professionals where it had been hard to describe what she experienced.

For Lars, engaging in treatment had provided a new and wider perspective of having a child with pain, and a shared language for him and his daughter.


*You’re more aware and attentive to the way and you have discussions about the choices she makes in life compared to the cost. You’ve found something of a common language to talk about certain things since you’ve already heard it to some extent you’ve got the tools in terms of thought processes to work with that persistent pain. (Lars, parent).*


Maria described how treatment had helped her and her husband to find new ways to communicate, as it had facilitated new perspectives on how to think about their situation and their parenting. *“You have different roles, that’s just how it is, um so, but it opened up a new dialogue for us and a new way of thinking, so we’ve really made changes there” (Maria, parent)*

Participation in ACT also resulted in improved communication between young people and their parents due to a developed understanding of the challenges of each individual’s unique situation. Julia described how she and her parent following treatment talked more about her pain, as treatment seemed to have increased her father’s interest in her experiences, prompting him to ask her more about how she feels and listening to her more attentively. *“I’ve told him all this before, it’s just I think he’s listening in a totally different way” (Julia, young person)*. Julia also noted how engagement in treatment helped her to appreciate how difficult having a daughter with ongoing pain and witnessing that pain was for her parents. “*Having a daughter who’s in pain, I mean you wouldn’t wish it on anybody” (Julia, young person)*

Perhaps most importantly, Anna described how engagement with treatment and other parents supported her to encourage her daughter to become more independent and live her own life. She described how treatment had helped her realize her daughter’s need for increased independence. 


*When you’ve got a sick child you become incredible overprotective er and that might not always be of benefit to your child, they do need to be allowed to live their own lives and do exciting things, and this other mum that I met, she let her daughter go inter-railing, with her sister I should add, but still, at that age, I thought was really cool, and it made me think, yeah, it makes sense. (Anna, parent)*


With her own words, she described this insight as *“sort of like cutting the umbilical cord” (Anna, parent)*, something which also seemed to imply a sense of freedom for herself. Anna had in the end let her own daughter travel too, and concluded:


*Perhaps I saw, pff what’s the word, us as two people, sometimes you can have symbiosis too, er, and it’s not, I can’t, it’s not mine, it becomes my pain in a sense because she is my child but I’m not the one who’s in pain and I have to help liberate her (Anna, parent)*


Decreasing parent protective behaviors is a common goal in pediatric chronic pain treatments, including ACT, and for ACT in specific, the overall goal is to increase parent psychological flexibility. As illustrated by Anna, the brief parent support provided as part of the ACT treatment facilitated both reduced protective parent behaviors, and increased her psychological flexibility, i.e., ability to notice difficult thoughts and feelings while choosing behaviors in line with her valued direction, i.e., supporting her daughter’s independence.

## 4. Discussion

In this study, the lived experiences of adolescents and parents who had participated in ACT for pediatric chronic pain were explored. Study findings resulted in the creation of three superordinate themes: *‘Warning system’ (theme 1)* centered around explanations for pain; *‘Change and challenges’ (theme 2)*, reflected both consistent and divergent participant experiences of treatment exercises; and finally, *‘A common language’ (theme 3)* that focused on the importance of communication. A starting point for psychological treatments for chronic pain is to motivate the patient to adopt a psychological approach to managing pain as an experience involving thoughts, feelings and behaviors. A clear rational is needed regarding the aim of treatment in order to move away from biomedical explanations and treatment approaches. To enable a shift in perspective, from symptom reduction to living well with pain, pain education is thought to be a crucial component. Pain education components are commonly included in many pediatric pain treatments, see, e.g., [20]. When they—as in this ACT treatment—include information about how the context informs pain perception, the neurophysiology of pain, and explanatory models such as the biopsychosocial model and the fear-avoidance model, they are broadly referred to as pain neuroscience education [53]. Studies examining the direct effect of providing primary pain neuroscience education to young people who experience chronic pain, or their parents, are currently lacking [53]. Given that some participants in this study experienced the pain education as revelatory, but others described problematic use of metaphors and difficulty understanding the information, the effects of pain education, and how it can be tailored, e.g., to reduce diagnostic uncertainty [7] or help unite parents and children in a shared coherent view of what the pain problem consists of and how it could be managed, warrants further research.

For current treatments for pediatric chronic pain, little is known about which specific treatment components are related to changes in important outcomes. In this study, the values component of the ACT treatment had been perceived as helpful for all, and participants seemed to have understood this part of treatment in the same way. From a contextual behavioral perspective, values are operationalized as verbal motivational antecedents with augmenting functions, i.e., they augment the reinforcing functions of consequences following values-consistent behavior [54]. Young people in this study described how talking about valued activities, writing them down, and visualizing doing them had served as a prompt and motivating antecedent for engaging in values-consistent behavior. Their parents also echoed the importance of values as a basis for behavior change. This suggests that exercises completed as part of the values work in ACT may be of particular importance. An example of such an exercise may be to first put in print what you value in life (e.g., friendship), then visualizing yourself doing the valued activity (e.g., spending time with your friends), talking about the feelings and thoughts that are related to those valued activities (sadness that you are currently not engaging in the activity, but also the happiness and fun it used to bring about), and visualize a concrete behavioral action (‘call my friend tomorrow, even if I don’t feel well’). From an RFT-perspective, these types of imaginary and experiential values exercises serve to augment stimulus functions and establish distal consequences in the present and thus further amplify the motivational functions of personal values, and potentially increase the efficacy of these specific interventions.

Interestingly, recent experimental research on mental imagery (i.e., ‘seeing in your mind’s eye’, ‘hearing in your mind’s ear’), suggests that engaging in motivational imagery may enhance the motivation for engagement in planned activities [55]. In light of the implied benefits of these exercises and that values work has not been studied in detail with adolescents [56], the direct effects of specific values interventions should be explored further with the pediatric chronic pain population.

Reflecting on other exercises included in treatment, participants’ experiences differed. Some found particular exercises helpful, while others held opposing views. In a recent study on treatment satisfaction after an intensive interdisciplinary pain treatment program for pediatric chronic pain, treatment methods were found to be the most frequent reason for both treatment satisfaction and dissatisfaction [57]. These authors suggest that further individual adjustment may thus be needed for some families. Our findings support and extend these findings by providing additional hypotheses for how such adaptations could be made. Young participants in this study appreciated having a variety of exercises to freely choose from. Also, they described that the aim or purpose of some exercises were unclear, and that some exercises and metaphors were developmentally inappropriate or too difficult. This implies the need to further consider how exercises in pain treatment programs are presented and ensure that the rationale for a specific exercise is sufficiently clear for the patient. In the ACT literature, as well as in pediatric pain literature [58], the importance of developmental consideration and adaptation and careful choice of metaphors for young people has been previously highlighted [56,59], but our findings show that further practical guidance and training, and examples on how to do such individual adaptation may be necessary for clinicians.

Although patient public involvement (PPI) appears central to improve services and research in regard to young people, barriers for pediatricians to engage in such involvement include lack of time, support and funding [60]. An enhanced focus on patient preferences, increased opportunities for decision making within the treatment, and involvement of patient representatives could be important steps in order to improve treatments for pediatric chronic pain. This view is also central to person-centered approaches that focus on the specific needs of patients and designs care in line with these, and involves patients in the planning, decision making, problem solving and goal setting related to these needs [61].

Experiences from communication and interaction with others emerged as a key theme in the present study. These experiences are not unique to ACT, and research points to the specific relevance of these factors in the context of medically unexplained symptoms [62] and chronic pain [63]. The importance of pain communication has received increased attention in recent years. For adults with chronic pain, experiences of validation and invalidation in the communication with their physician seem to be of specific importance for patients with high pain interference and negative affectivity [64]. However, mere validation, alliance building and information about the biopsychosocial model had no significant effect on outcomes for adults with chronic pain [65]. In-depth analysis of the communication that takes place during ACT, for example, in conjunction with an observational study, like one that focused on the behaviors of parents of young people with chronic pain who were undergoing residential pain treatment [66], could shed further light on pain communication for young people, in their interaction with one another, as well as with their family and their treatment provider.

Although a sense of shared understanding and learning that others struggle with the same experiences can arise from group-based psychological interventions in general, as well as ACT-specific programs like a school-based intervention targeting depression and stress [67], and be perceived as helpful, the opposite may also occur. One participant in this study described how she felt sad and hurt from comments from a fellow group member, which resulted in her feeling alienated from the others. Another participant mentioned feeling insulted in session by the treatment provider. Such experiences are important to learn from and should be subject for further study as they may be potent reasons for attrition and unsuccessful treatment outcomes. Psychological treatments may cause unwanted or adverse events, such as negative wellbeing and worsening of symptoms [68], and monitoring of adverse events is highly encouraged in future studies with the pediatric chronic pain population [19].

The lived experiences of parents in the present study largely mirrored those of the young persons. Prior research exploring parent experiences in relation to treatment in the pediatric chronic pain context have shown the importance of helping parents deal with their own distress, in order to effectively support their child, see e.g., [69,70], but the principles they are taught in treatment may not always resonate with them [71]. Our results highlight the need for parents to understand the rationale for treatment and be on board with adopting a psychological approach, and the need to tailor treatment according to specific parent and family needs. One important treatment outcome for parents who are involved in treatment for pediatric chronic pain is decreased protective behaviors. In the present study, participants linked the changes they had noted in such protective behavior with both the values component and the interaction with fellow parents. Future research using other research designs could explore whether adding a specific values-component would be beneficial in other support programs currently in use for parents of young people with chronic pain [72] and whether changes in parent protectiveness can be attributed to values-based work specifically. Targeting parent psychological flexibility has been suggested to be of importance for increasing both child and parent functioning [26,32,73,74,75,76]. The present study extends these prior quantitative findings by adding parent descriptions of how some parents describe increased psychological flexibility, i.e., how treatment helped them engage in long-term effective behaviors to support their children in the presence of their own worry and distress.

A few limitations should be taken into consideration when interpreting and discussing the results. During the study, the hospital underwent a major reorganization in which the clinic relocated and experienced staff turnover. This delayed the onset of treatments and scheduling of follow-up appointments, which in turn, affected recruitment and prolonged the data collection period. Further, member check was attempted by inviting participants per post to comment on a brief and accessible summary of preliminary results. As only two did so, this suggests the need for a more active approach, e.g., phoning participants, or scheduling a second interview in advance. As commonly seen in studies with the pediatric chronic pain population, the majority of participants are female; therefore, subsequent work in this area should include specific strategies to recruit both young males with chronic pain and fathers. Finally, in accordance with the qualitative focus of the study, we did not include evaluations of participants’ response to treatment or their ratings of treatment satisfaction. Thus, we do not know if the young people and parents who participated were all treatment responders or particularly satisfied with the treatment, i.e., a biased sample. However, as our results cover a variety of experiences from treatment, we conclude the risk of such bias to be low.

Strengths of this study include the combined perspectives of young people and their parents. We used an idiographic approach, which allows the diversity in lived experiences to be reflected. Clinical implications include the need for tailored treatment according to patient preferences and needs and to invite participants into such decisions around treatment content when possible. Directions for future research include evaluating the specific effects of pain education, values-based exercises, and pain communication strategies for young people who experience chronic pain and their parents, but also how these components relate to functional outcomes and core constructs, i.e., psychological flexibility, in ACT for pediatric chronic pain.

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
