# Peer review of "Adolescent and Parent Experiences of Acceptance and Commitment Therapy for Pediatric Chronic Pain: An Interpretative Phenomenological Analysis"

_children, 2019, doi:10.3390/children6090101_

Round 1

Reviewer 1 Report

In this study, the lived experiences of adolescents and parents who had participated in an ACT-based interdisciplinary treatment for pediatric chronic pain were explored. I am grateful to read a study that allowed to present perspectives of pediatric patients with chronic pain and their caregivers. Even though it is a very small sample, the content is very valuable and differs from most current studies that focus on quantitative and standardized measures. I  like to suggest the following:

to switch the characterstics of the participants (age, symptom duration) to the result part and add further characteristics such as school abscence and withdrawal from physical activity to demonstrate disease severity. It may be helpful to know wether parents have a history of a pain condition as it may influence their perspectives. to add the information that the participants agreed to have their quotes published and that their names could be used (or were changed) - just for clarification purposes- I am aware that ethics approval was received

Author Response

Reply to reviewer 1.

COMMENT: Comments and Suggestions for Authors

In this study, the lived experiences of adolescents and parents who had participated in an ACT-based interdisciplinary treatment for pediatric chronic pain were explored. I am grateful to read a study that allowed to present perspectives of pediatric patients with chronic pain and their caregivers. Even though it is a very small sample, the content is very valuable and differs from most current studies that focus on quantitative and standardized measures.

REPLY: Thank you very much for this positive summary of the manuscript and for your helpful suggestions, which we are happy to respond to point-by-point below.

COMMENT: I like to suggest the following: To switch the characterstics of the participants (age, symptom duration) to the result part and add further characteristics such as school abscence and withdrawal from physical activity to demonstrate disease severity.

REPLY: Thank you very much for suggesting that we expand the participant characteristics to include examples of pre-treatment pain related dysfunction. We agree that this is an important addition, which improves the understanding of how pain influenced the lives of these young people and their parents. These added sections now reads (p.3, line 125 and onwards):

   “Examples of pain-related impairment in functioning prior to treatment included limitations in physical activity, extensive school absence (e.g. only present 50% over the course of a year, or staying at home full-time for months at a time), pain episodes that included fear of dying or being unable to walk, numerous health-care visits, and emotional and social impairment (e.g. depressive symptoms, fear of pain, and feeling isolated and different from others).”

    “Parents described pain-related family dysfunction and difficulties in parenting prior to treatment. These included major adaptations in family activities and routines to reduce the child’s pain, constantly being on call for their child, and feeling guilty when they had encouraged everyday activities or initiated medical examinations that provoked pain.”

However, regarding the reviewer´s suggestion to move participant characteristics to the results section, we would prefer to keep this information as part of the method section. In IPA, ideographic detail is key for understanding the subsequent results, and participant characteristics are thus commonly placed in the methods-section as they do not constitute results in that sense (see e.g. recommendations by Smith, Flowers and Larkin [1]). Should the reviewer still prefer that we move these descriptions to results, we are of course happy to reconsider our decision.

COMMENT: It may be helpful to know wether parents have a history of a pain condition as it may influence their perspectives.

REPLY: Thank you, we agree that presence of parent chronic pain may be an important aspect. Unfortunately, we did not collect this data as part of this study. We have added this information to the participant characteristics section for clarity, and it reads (p. 3, line 131 and onwards):

   “No other parent characteristics (e.g. presence of parent chronic pain) were collected in this study”.

COMMENT: To add the information that the participants agreed to have their quotes published and that their names could be used (or were changed) - just for clarification purposes- I am aware that ethics approval was received.

REPLY: We agree that it is important to be very clear about how participants were informed about the details of the study, and we have clarified this further in the manuscript. The added sentences reads (p. 3, line 112 and onwards):

“After this phone call, those considering study participation received detailed written study information together with consent forms per post. This included information about that any quotes used from participants would be presented in a way that would protect their anonymity.”

References

Smith, J., P. Flowers, and M. Larkin, Interpretative phenomenological analysis. Theory, method and research. 2009, Los Angeles: SAGE.

Reviewer 2 Report

This manuscript reports a qualitative analysis of four youth participating in an 18-session ACT-based psychological treatment as well as their four parents. Three youth were treated in a group format, whereas the other was treated individually, but all received the same treatment components. While the sample size is small, it is reported as acceptable for the type of analysis chosen (IPA), which does not require saturation of identified themes. The manuscript describes the process of interview and analysis with good detail, and while I have limited experience with qualitative analyses I was able to understand the process sufficiently to understand how the team reached the presented results. The results themselves are of interest, and go beyond simple validations of what we already do. These results suggest important next steps such as needing to tailor metaphors, allowing some choice with the exercises, targeting parents’ own values, and being sensitive to parents who might not want to feel like the target for the treatment. Further, results helpfully identified some specific advantages and disadvantages to the group-based treatment. The discussion is lengthy, but brings up a number of important points that largely tie back to the results of the study.

I was confused as the treatment is described as interdisciplinary, but there is limited information about any treatment component beyond psychological. Thus, while the aim of the analysis is to focus on ACT-related constructs, it is hard to say to what extent the themes reflect response to interdisciplinary treatment more broadly. I have also identified some areas where providing more information would be helpful, and a few other minor suggested changes. All requested changes are detailed below. With these changes I believe the manuscript would provide a strong contribution to the field.

Concerns and suggestions:

Introduction (p.2, line 49). Given the mention of intensive treatment here, and the term “interdisciplinary” in the manuscript title, I found myself anticipating that this study would have been conducted in the setting of an intensive program. Consider rewording to more clearly get at your point here which is the need for understanding mechanisms related to youth and parents, e.g. “While parent support is commonly included in evidence-based treatments [19,20], more research is needed to improve…” In the aims (p.2, line 80), be more specific about the ACT-specific direction of the investigation. Consider incorporating some of the language from the data collection section (e.g. p.4, line 149). Regarding the interdisciplinary treatment, please note if PT is also included, describe any medical evaluation and medication treatment, and if possible outline content of education sessions. This could be expanded from what is currently in the participants section, and perhaps moved to the treatment description section. The fact that some received individual and some group treatment is somewhat buried by its location in the participants section. Consider more specifically indicating this in the treatment description section.

Author Response

Reply to reviewer 2.

COMMENT: This manuscript reports a qualitative analysis of four youth participating in an 18-session ACT-based psychological treatment as well as their four parents. Three youth were treated in a group format, whereas the other was treated individually, but all received the same treatment components. While the sample size is small, it is reported as acceptable for the type of analysis chosen (IPA), which does not require saturation of identified themes. The manuscript describes the process of interview and analysis with good detail, and while I have limited experience with qualitative analyses I was able to understand the process sufficiently to understand how the team reached the presented results. The results themselves are of interest, and go beyond simple validations of what we already do. These results suggest important next steps such as needing to tailor metaphors, allowing some choice with the exercises, targeting parents’ own values, and being sensitive to parents who might not want to feel like the target for the treatment. Further, results helpfully identified some specific advantages and disadvantages to the group-based treatment. The discussion is lengthy, but brings up a number of important points that largely tie back to the results of the study.

REPLY: Thank you very much for this positive summary of the manuscript.

COMMENT: I was confused as the treatment is described as interdisciplinary, but there is limited information about any treatment component beyond psychological. Thus, while the aim of the analysis is to focus on ACT-related constructs, it is hard to say to what extent the themes reflect response to interdisciplinary treatment more broadly. I have also identified some areas where providing more information would be helpful, and a few other minor suggested changes. All requested changes are detailed below. With these changes I believe the manuscript would provide a strong contribution to the field.

REPLY: Thank you for these clear and helpful suggestions, we are delighted to respond to them point-by-point below.

COMMENT: Concerns and suggestions:

Introduction (p.2, line 49). Given the mention of intensive treatment here, and the term “interdisciplinary” in the manuscript title, I found myself anticipating that this study would have been conducted in the setting of an intensive program. Consider rewording to more clearly get at your point here which is the need for understanding mechanisms related to youth and parents, e.g. “While parent support is commonly included in evidence-based treatments [19,20], more research is needed to improve…”

REPLY: We appreciate this concern, and agree that clarification is needed to avoid confusion. It also relates to the point raised below, regarding aspects of the interdisciplinary nature of the treatment.

Though the ACT-based treatment at the clinic where this study was conducted can include sessions with physiotherapists as well as psychologists, physicians and nurse, the participants in this study did not receive PT sessions. Though both psychologists, physicians and a nurse were involved in delivering the treatment to these participants, we agree that most of the program consisted of psychological components. We have decided to revise the title and how we refer to the treatment throughout the manuscript. The revised title now reads:

“Adolescent and Parent Experiences of Acceptance and Commitment Therapy for Pediatric Chronic Pain: An Interpretative Phenomenological Analysis”

Further, we have expanded the sentence on the need to understand the active ingredients in treatment, and following suggestions from reviewer #3 we have also included information about the rate of non-responders after these evidence-based therapies. The revised section (p. 2, line 51 and onwards) now reads:

“Research supports the use of face-to-face psychological therapies and interdisciplinary intensive treatment to reduce pain and restore function in pediatric chronic pain [19, 20] and parent support is commonly included in treatment, with beneficial effects [21]. However, these evidence-based treatments are not effective for all children, and some domains of pain-related dysfunction generally does not improve. For example, two studies report that 22-27% of children show no response after intensive interdisciplinary treatment [22, 23], and generally, positive effects on depression and anxiety are lacking for psychological therapies [19]). A clearer understanding of the processes and mechanisms that are related to successful outcomes for all domains of functioning, and the active ingredients of treatment, would advance our understanding of how to improve treatment outcomes [19].”

COMMENT: In the aims (p.2, line 80), be more specific about the ACT-specific direction of the investigation. Consider incorporating some of the language from the data collection section (e.g. p.4, line 149).

REPLY: Thank you for this suggestion. Though the treatment that was delivered was an ACT treatment, we did not pre-specify that ACT-related constructs would be described by participants (i.e. it may have been other things than ACT-specific interventions, e.g. treatment components included in most psychological therapies and/or interdisciplinary programs for pediatric chronic pain) that were perceived as helpful by young people and their parents). We agree that this section needs clarification, and we have revised as shown below (p. 2, line 85 and onwards): 

“Further, as described above, a few studies indicate that central treatment targets in ACT (e.g. acceptance) are related to improvements in outcomes for young people with chronic pain. However, no study has yet to our knowledge applied qualitative methods to explore participant experiences of ACT-treatment for pediatric chronic pain, and what may have been perceived as helpful to achieve change for participants (e.g. do participant experiences include or relate to the explicit targets of ACT, and the processes thought to facilitate change). Thus, the aim of this cross-sectional qualitative study was to explore the lived experiences of young people and parents with regard to participating in ACT for pediatric chronic pain.”

COMMENT: Regarding the interdisciplinary treatment, please note if PT is also included, describe any medical evaluation and medication treatment, and if possible outline content of education sessions. This could be expanded from what is currently in the participants section, and perhaps moved to the treatment description section.

REPLY: Thank you for pointing this out. We have clarified this information according to your suggestions. This section in the treatment description now reads (p. 3, line 109 and onwards):

“Both parents and children received pain education sessions, delivered by a physician or a nurse. These sessions included descriptions of the pain system, the differences between acute and chronic pain, and of how pain perception is context dependent, with the aim of facilitating a shift in perspective from symptom reduction to acceptance of pain and valued living. Specific medical evaluations or adjustments in medical treatments, along with physiotherapy sessions, were available as treatment add-ons if needed.”

We have also clarified that the young people in this study did not receive any additional PT or medical sessions. (p. 4, line 159):

  “They did not have additional physiotherapy or medical evaluation sessions.”

COMMENT: The fact that some received individual and some group treatment is somewhat buried by its location in the participants section. Consider more specifically indicating this in the treatment description section.

REPLY: We agree, and we have added this information to the treatment description as well. It now reads (p. 3, line 98):

      “The treatment could be delivered either as individual treatment or in a group format.”

Reviewer 3 Report

Thank you for the opportunity to review this well-written, novel qualitative examination of adolescent and parent experiences of ACT for chronic pain. Given the importance of considering the lived experiences of patients, particularly in the context of tailoring patient-centered treatments, this study is a valuable contribution to the growing literature examining the utility of ACT-based approaches for treating pediatric chronic pain. Please see recommendations for minor revisions and additional comments below:

Page 2, Paragraph 2—The reader would benefit from an additional statement indicating treatment response rates of psychological/interdisciplinary interventions for pediatric chronic pain. Doing so would provide context/support for the statement on line 51 indicating that more research is needed to improve treatment outcomes.

Page 3, Paragraph 2—Inclusion criteria included “recent participation” in treatment. Please provide additional detail indicating how long ago participants were in treatment. This may be relevant given perspective changes of their experience over time.

Do the authors have quantitative measures of treatment outcomes or treatment satisfaction for the participants in this study? I would be concerned that participants included in the study were self-selected “satisfied” participants from the overall sample of youth who were engaged in treatment. If this information is not available, please acknowledge the possibility of this bias in the limitations.

Nicely organized and presented Results section and very thoughtful, clinically relevant Discussion section.

Author Response

Reply to reviewer 3.

COMMENT: Thank you for the opportunity to review this well-written, novel qualitative examination of adolescent and parent experiences of ACT for chronic pain. Given the importance of considering the lived experiences of patients, particularly in the context of tailoring patient-centered treatments, this study is a valuable contribution to the growing literature examining the utility of ACT-based approaches for treating pediatric chronic pain. Please see recommendations for minor revisions and additional comments below:

REPLY: Thank you very much for this positive summary of the manuscript. We are very happy to respond to these recommendations point-by-point below.

COMMENT: Page 2, Paragraph 2—The reader would benefit from an additional statement indicating treatment response rates of psychological/interdisciplinary interventions for pediatric chronic pain. Doing so would provide context/support for the statement on line 51 indicating that more research is needed to improve treatment outcomes.

REPLY: Thank you, we agree and we have added this information with relevant references, and also revised this section slightly following suggestions from reviewer #2. The section now reads (p. 2, line 49 and onwards):

“Research supports the use of face-to-face psychological therapies and interdisciplinary intensive treatment to reduce pain and restore function in pediatric chronic pain [19, 20] and parent support is commonly included in  treatment, with beneficial effects [21]. However, these evidence-based treatments are not effective for all children, and some domains of pain-related dysfunction generally does not improve. For example, two studies report that 22-27% of children show no response after intensive interdisciplinary treatment [22, 23], and generally, positive effects on depression and anxiety are lacking for psychological therapies [19]). A clearer understanding of the processes and mechanisms that are related to successful outcomes for all domains of functioning, and the active ingredients of treatment, would advance our understanding of how to improve treatment outcomes [19].”

COMMENT: Page 3, Paragraph 2—Inclusion criteria included “recent participation” in treatment. Please provide additional detail indicating how long ago participants were in treatment. This may be relevant given perspective changes of their experience over time.

REPLY: We agree that this is relevant, and we have added the mean time since end of treatment for the participants. It reads (p. 4, line 162):

“The mean time between end of treatment and their participation in the interview was 11.8 weeks (Weeksmin –max: 8.3-22.1).“

COMMENT: Do the authors have quantitative measures of treatment outcomes or treatment satisfaction for the participants in this study? I would be concerned that participants included in the study were self-selected “satisfied” participants from the overall sample of youth who were engaged in treatment. If this information is not available, please acknowledge the possibility of this bias in the limitations.

REPLY: We understand the reviewer’s concern. As this is solely a qualitative paper, we did not include quantitative measures of treatment outcome or satisfaction in this study, but we agree that this is an important aspect. We did however take great care to prompt clinicians to ask both ‘satisfied’ patients as well as those who were not, if they were interested in receiving information about participation in the study. We have added a sentence on this in the manuscript, and it reads (p. 3, line 127):

       “Clinicians had been prompted by the researchers to ask both participants who they thought had responded well to treatment and were satisfied, and those who were not.”

We have also acknowledged the possibility of bias in the limitation section. This sentence now reads (p. 14, line 701):

“Finally, in accordance with the qualitative focus of the study, we did not include evaluations of participants’ response to treatment or their ratings of treatment satisfaction. Thus, we do not know if the young people and parents who participated were all treatment responders or particularly satisfied with the treatment, i.e. a biased sample. However, as our results cover a variety of experiences from treatment, we conclude the risk of such bias to be low.”

COMMENT: Nicely organized and presented Results section and very thoughtful, clinically relevant Discussion section.

REPLY: Thank you very much, we appreciate this positive feedback.
